# MicroRNAomic Analysis of Spent Media from Slow- and Fast-Growing Bovine Embryos Reveal Distinct Differences

**DOI:** 10.3390/ani14162331

**Published:** 2024-08-13

**Authors:** Paul Del Rio, Sierra DiMarco, Pavneesh Madan

**Affiliations:** Department of Biomedical Sciences, Ontario Veterinary College, University of Guelph, 50 Stone Road East, Guelph, ON N1G 2W1, Canada

**Keywords:** miRNA, IVF, biomarker, embryonic health, pre-implantation, embryo development

## Abstract

**Simple Summary:**

Embryos release microRNAs (miRNAs) into their surrounding spent media during early development. Gene expression is expected to vary between embryos of good and poor developmental potential; therefore, identifying miRNAs unique to good-quality embryos can aid in embryo selection. This study aimed to characterize the miRNA expression in the spent media of embryos identified as slow- versus fast-growing, as developmental timing may be indicative of embryo quality. Distinct miRNA populations were detected in the spent media conditioned with bovine embryos growing at different developmental rates at the two-cell, eight-cell, and blastocyst stage in vitro. The results highlight the novel and non-invasive miRNA biomarkers of early embryo development.

**Abstract:**

In bovine embryos, the microRNA (miRNA) expression has been profiled at each stage of early development in vitro. The miRNAomic analysis of spent media has the potential to reveal characteristics of embryo health; however, applications are limited without categorizing miRNA profiles by embryo quality. Time-lapse imaging has shown the timing of embryo development in vitro may be indicative of their developmental potential. The study aimed to characterize miRNAs in the spent media of bovine embryos with different growth rates during the pre-implantation phase. Bovine cumulus–oocyte complexes were aspirated from ovaries, fertilized, and cultured to blastocyst stage of development. At the 2-cell, 8-cell, and blastocyst stage, each microdrop of 30 presumptive zygotes were classified as slow- or fast-growing based on the percentage of embryos that had reached the desired morphological stage. A comparative analysis was performed on the spent media of slow- and fast-growing embryos using the results of a GeneChip miRNA 4.0 array hybridization. In total, 34 differentially expressed miRNAs were identified between the comparison groups: 14 miRNAs were found in the 2-cell samples, 7 in the 8-cell samples, and 12 in the blastocyst samples. The results demonstrate distinct miRNAs populations can be identified between slow- and fast-growing embryos, highlighting the novel biomarkers of developmental potential at each stage of pre-implantation development.

## 1. Introduction

It is well-established that embryos of differing developmental potential have different genomic, proteomic, and metabolomic profiles [1]. These variations in expression are detectable intracellularly, and, more recently, in the spent culture media of in vitro fertilization (IVF) systems [1,2]. This presents an opportunity to optimize the embryo selection process and improve prognosis, as selection is crucial for successful implantation and live birth rates [3]. The traditional morphological evaluation is still considered the gold standard for embryo selection [4,5]. Cleavage rates and blastocyst formation are often judged subjectively, so some selected embryos may fail to implant or miscarry from underlying chromosomal abnormalities [6]. While pre-implantation genetic screening (PGS) can detect aneuploidies, this service requires an invasive embryo biopsy and can be costly [7]. Therefore, there is a need to establish a non-invasive, highly accurate method of embryo selection. The analysis of small molecules found in the spent media (SM) has the possibility of revealing biomarkers related to intrinsic embryo physiology. Strong evidence suggests that array-based metabolomics and miRNAomic analysis are good candidates for adjunct assessment methods of embryo quality. Metabolomics and the miRNAomic profiling of embryo SM have revealed distinct signatures between embryos of different morphological appearance [8], chromosomal status [9], sex [10], and implantation outcome.

Advancements in time-lapse imaging have also revealed the importance of a morpho-kinetic assessment in assessing embryo quality [11,12]. Time-lapse imaging allows the embryologist to visualize the embryos in the incubator in between assessments, while minimizing the negative effects of removing the embryos from the incubator. This is an important tool, given the status and grade of an embryo can change within hours [13,14]. This relatively new technique has shown the timing of the onset and duration of key morphological events, such as cleavage, compaction, and blastocyst formation, may indicate normal and aberrant embryo development. Since in vivo embryos develop faster than their in vitro counterparts, it is commonly accepted that faster-developing in vitro embryos are healthier [15]. In fact, research has shown in vitro embryos that cleave earlier have higher blastocyst rates. The timing may be indicative of the stress experienced by an embryo. The absence or low levels of stress factors such as reactive oxygen species may mean embryos can develop faster as less time can be spent initiating repair pathways [15].

However, other groups have presented a counter idea that the slower-growing embryo has more time to correctly initiate and choreograph the events of embryogenesis. Research by Market-Velker and colleagues compared slow-growing (SG) and fast-growing (FG) embryos with in vivo embryos on factors such as methylation status, expression of imprinted genes, embryo cell number, and morphology [16]. Their findings showed that SG embryos were most similar to in vivo embryos on all parameters measured [15]. Genomic imprinting and the expression of metabolic markers in SG embryos closely mirrored those of in vivo embryos. Market-Velker and colleagues postulate FG embryos may transition too rapidly during the first few embryonic stages, causing an inability to maintain epigenetic information [16]. Thus, it appears embryos grow within a time range, whereby anything too slow may indicate a pathological condition, while anything too fast may signify an embryo erroneously rewired to move onto the next developmental stage.

To date, few studies have profiled SM conditioned with SG and FG embryos. One study that did so examined the metabolites present in the SM cultured with SG and FG embryos. Using nuclear magnetic resonance (NMR), Perkel and Madan were able to detect distinct metabolic signatures in the SM between SG and FG embryos at the 2-cell, 8-cell, 16-cell, and blastocyst stage of development [17]. Specifically, their data showed distinct differences between media cultured with the 4-cell SG and FG embryos for pyruvate, and at the 16-cell stage for acetate, tryptophan, leucine/isoleucine, valine, and histidine. Four-cell SG embryos had a higher consumption of pyruvate, while 16-cell SG embryos released more acetate into the media, in comparison to their FG counterparts [17]. Acetate is produced when there is an over-abundance of acetyl-CoA, such as in situations where the Krebs cycle or electron-transport chain is malfunctioning. Since both processes lie within the mitochondria, this observation suggests SG embryos exhibit some level of mitochondrial dysfunction, resulting in a metabolic disturbance.

The embryonic mitochondrial biogenesis analysis conducted in our lab revealed metabolic distress may be related to mitochondrial dysfunction. In this study, GLYCOX and OXPHOS gene expression were examined in SG and FG embryos at the 2-cell, 8-cell, morula, and blastocyst stage of development [18]. GLYCOX and OXPHOS genes regulate the pathways in embryo energy production, whereby OXPHOS is a mitochondrial-dependent pathway and dominates in early embryo development, and GLYCOX dominates in late embryo development [18]. The data indicate SG embryos had a higher expression of both OXPHOS and GLYCOX genes at all time-stages in comparison to FG embryos. This over-expression may serve as a compensatory mechanism for mitochondrial dysfunction occurring in SG embryos. Overall, the data from our lab suggest that metabolic assays are sensitive in detecting differences between embryos growing at different rates [18].

Recent data from the Madan lab has also shown the miRNA expression can be detected in the SM of embryos cultured in vitro [19]. This study clearly demonstrated that SM can be used to detect the miRNA expression from embryos harvested at timed stages of developmental. However, determining the global SM miRNA expression profiles between SG and FG embryos has not been attempted. Therefore, the objective of the present study was to globally profile, using a heterologous miRNA microarray, the miRNAs expression in the SM of SG and FG embryos at the 2-cell, 8-cell, and blastocyst stage of development.

## 2. Materials and Methods

### 2.1. Ethics

All experiments were conducted in accordance with the requirements of the Animals for Research Act of Ontario, and the Canadian Council for Animal Care (CCAC) [20]. No authorization was required as the work was carried out on ovaries collected from slaughterhouse.

### 2.2. Chemicals

All chemicals and media were obtained from Sigma-Aldrich (Oakville, ON, Canada) unless otherwise specified.

### 2.3. Bovine Oocyte Collection and In Vitro Maturation

Bovine ovaries were sourced from a local abattoir (Cargill Canada, Guelph, ON, Canada) and transported to the laboratory under phosphate-buffered saline (PBS) containing NaCl (136.9 mM), Na_2_HPO_4_ (8.1 mM), KCL (1.47 mM), KH_2_PO_4_ (1.19 mM), and MgCl2.6H_2_O (0.49 mM), maintained at 35–36 °C. Aspirated 4–8 mm follicles were placed in HEPES-buffered Hams F-10 medium supplemented with 2% donor calf serum (PAA Laboratories Inc., Toronto, ON, Canada). In vitro bovine oocyte maturation was carried out as previously described [17,21]. High-quality cumulus–oocyte complexes (COCs) were washed three times—twice with 3 mL synthetic S-IVM (Sigma-Aldrich, Oakville, ON, Canada) and once with 3 mL S-IVM, 0.5 g/mL of follicle stimulating hormone, 1 g/mL of luteinizing hormone, and 1 g/mL of estradiol (Sigma-Aldrich, Canada). COC maturation took place in 80 μL drops of S-IVM under silicone oil for 22–24 h at 38.5 °C in an atmosphere of 5% CO2 with 100% humidity. Groups of 15–20 COCs with dark, homogenous cytoplasm surrounded by packed layers of granulosa cells were selected for maturation. Following maturation, approximately 20 COCs were washed four times—twice with 3 mL HEPES/Sperm TALP and twice with 3 mL IVF-TALP, then placed in 80 μL drops of IVF-TALP under a layer of silicone oil. HEPES/Sperm TALP consisted of HEPES buffered Tyrode’s albumin–lactate–pyruvate medium, with 15% BSA (0.0084 mg/mL final; fatty acid free, Sigma-Aldrich, Canada). IVF-TALP consisted of Tyrode’s solution with 15% BSA and 2 mg/mL heparin (Sigma-Aldrich, Canada).

### 2.4. In Vitro Fertilization of Bovine Embryos

As per the swim-up technique, thawed bovine sperm was placed in HEPES/Sperm TALP and incubated for 45 min at 38.5 °C in an atmosphere of 5% CO_2_ with 100% humidity before being centrifuged at 200× *g* for 7 min [22]. In vitro fertilization was carried out as previously described [17,21]. Briefly, to achieve fertilization, the COCs and sperm were co-incubated at a concentration of 1.0 × 10^6^ at 38.5 °C in 5% CO_2_ with maximum humidity. Then, 18 h post fertilization (hpf), the presumptive zygotes (PZs) were denuded to strip remaining cumulus cells, washed twice with 3 mL HEPES/Sperm TALP, and once with in vitro culture (IVC) media. IVC media contained CaCl_2_•2H_2_O (1.17 mM), KCL (7.16 mM), KH_2_PO_4_ (1.19 mM), MgCl_2_•6H_2_O (0.49 mM), NaCl (107.7 mM), NaHCO_3_ (25.07 mM), and Na lactate (60% syrup, 3.3 mM) (ChemiconMillipore, Billerica, MA, USA). The IVC medium was further supplemented with 50 μL of 100× non-essential amino acids (M7145), 100 μL 50x essential amino acids (M5550), 25 μL of sodium pyruvate (0.00886 mg/mL final), 2.5 μL of gentamicin (25 mg/mL final) (Invitrogen, Burlington, ON, Canada), and 280 μL of 15% BSA (0.0084 mg/mL final). PZs with homogenous cytoplasm were transferred into 6, 30 μL drops of IVC media under a layer of silicone oil at 38.5 °C in an atmosphere of 5% CO_2_, 5% O_2_, and 90% N_2_, in groups of 30 PZs per drop. It was made sure that no cumulus cells transferred into the in vitro culture drops to prevent any contamination of miRNA.

### 2.5. Collection of Spent In Vitro Culture Media Conditioned with SG and FG Embryos

Collection of spent media conditioned with SG and FG embryos is outlined in Figure 1. Specifically, on day 0 of culture, each of the 6 micro drops containing 30 PZ was assigned group numbers ranging from 1–6. This made it possible to follow the same cohort of embryos throughout the pre-implantation period while collecting SM at specific time points corresponding to the 2-cell, 8-cell, and blastocyst stage of development. Microdrops were classified as an SG or FG group at each time point, based on the percentage of embryos that have reached the desired morphological stage at a given time point [17,23]. At 18–30 hpf, microdrops were considered SG if the cohort had <50% reached the 2-cell-stage and FG if the cohort had ≥50% reached the 2-cell stage. Once the groups were designated SG or FG, the embryos were placed into fresh microdrops of IVC media, retaining their original group number, and placed in the incubator for another 30 h to reach the 8-cell stage. Approximately 25 μL of conditioned media from each microdrop from 2-cell SG and FG groups were collected, pooled, and placed in separate Eppendorf tubes. At the 8-cell stage, the cohort of embryos was assessed for 8-cell rate formation and cohorts were considered SG if the 8-cell rate was <50% and FG if the 8-cell rate was ≥50%. After designation, the embryos were placed into a fresh microdrop of IVC media, retaining their original group number, and placed in the incubator for another 72 h to allow for blastocyst formation. Approximately 25 μL of conditioned media from each microdrop from 8-cell SG and FG groups were collected, pooled, and placed in separate Eppendorf tubes. At the blastocyst stage, the cohort of embryos was assessed for a final time for blastocyst rate formation and cohorts were considered SG if blastocyst rates were <20% and FG if blastocyst rates were ≥20%. After designation, the embryos were taken out of the drop and the SG and FG embryos were placed in separate Eppendorf tubes and flashed frozen. Approximately 25 μL of conditioned media from each microdrop from SG and FG groups were collected, pooled, and placed separately in Eppendorf tubes. Eight IVF runs were completed until 1100 μL of SM was collected for each developmental stage: 2-cell SG/2-cell FG, 8-cell SG/8-cell FG, and blastocyst SG/blastocyst SG.

### 2.6. miRNA Extraction

miRNA was isolated from spent and unconditioned IVC media using the RNeasy mini kit (Qiagen, Hilden, Germany) according to the manufacturer’s protocol and as previously described [19]. A total RNA sample was required for downstream array analysis. Briefly, 350 μL of spent and unconditioned IVC media was transferred into a 2.5 mL Eppendorf tube, followed by the additional of an equal volume of QIZzol lysis reagent. The mixture was vortexed for 20 s and allowed to sit at room temperature for 10 min. Subsequently, 350 μL of chloroform was added. The mixture was incubated for 2 min at room temperature and centrifuged at 12× *g* (15,000 RPM) at 4 °C for 15 min. To achieve total RNA separation, the resulting supernatant was placed into the RNeasy MinElute spin column. Subsequent washing steps using buffer RWT, buffer RPE, and 80% ethanol were completed. Total RNA was eluted using 30 μL of RNAse-free water and stored at 80 °C prior to microarray analysis. In total, 3 biological replicates of pooled SM from 2-cell SG/2-cell FG, 8-cell SG/8-cell FG, and blastocyst SG/blastocyst FG groups and plain media were processed and prepared for microarray analysis.

### 2.7. miRNA Microarray Hybridization

Microarray processing was performed by our colleagues at Genome Quebec (McGill University, Montreal, QC, Canada). The microarray profiling was using the Affymetrix GeneChip miRNA 4.0 assay (Affymetrix, Santa Clara, CA, USA), according to manufacturer’s instructions and as described previously by Reza et al., 2018 [24]. In brief, RNA samples were labeled with the FlashTag Biotin RNA Labelling Kit (Genisphere, Hatfield, PA, USA), quantified, fractionated, and hybridized to the miRNA microarray. The procedure involved heating the labeled RNA to 99 °C for 5 min, then at 45 °C for 5 min, followed by hybridization with constant agitation at 60 rpm for 16 h at 48 °C on an Affymetrix 450 Fluidics Station (Bedford, MA, USA). The microarray chip was then washed and stained with Genechip Fluidics Station 450 (Thermofischer Scientific, Waltham, MA, USA), then scanned with an Affymetrix GCS 3000 scanner (Bedford, MA, USA). The data were processed using the Affymetrix Genechip command console software version 7.2 (Thermofischer Scientific, Mississauga, ON, Canada).

### 2.8. Statistical Analysis

CEL files were imported in RMA + DMG (all organisms) mode for Genechip microarray analysis using the Transcriptome Analysis Console^®^ 4.0.2.15 (TAC) software. Employing fold-change and an independent T-test, a comparative analysis was performed between the SM samples 2-cell SG/2-cell FG, 8-cell SG/8-cell FG, and blastocyst SG/blastocyst FG and the control (unconditioned media). The null hypothesis assumed no difference between the 2 groups. Probe-sets were deemed expressed if ≥50% of samples had a false discovery rate (FDR) <0.05 and detectable above background (DABG) values below DABG threshold of <0.05. Probes were considered differentially expressed if they exhibited a fold-change of ≤−2 or ≥2 (*p*-value < 0.05). The TAC software (version 4.0.2.15) was used for all statistical analyses and the display of differentially expressed genes. 

### 2.9. mRNA Target Pathway Prediction of Differentially Expressed miRNAs

A gene-list was constructed using TargetScan Human 7.2 (http://www.targetscan.org/vert_72/) under Cow annotation to perform a functional analysis of the differentially expressed miRNAs (DEM) found between 2-cell SG/2-cell FG, 8-cell SG/8-cell FG, and blastocyst SG/blastocyst FG SM. The list contained genes with a total context score of <−0.5. Gene-set enrichment analysis (GSEA) was performed on the gene-list, using DAVID 6.8 (https://david.ncifcrf.gov/), selecting the gene-ontology: biological processes option. A *p*-value of <0.05 indicated a pathway was significantly enriched.

## 3. Results

### 3.1. Differentially Expressed miRNAs between 2-Cell SG vs. 2-Cell FG, 8-Cell SG vs. 8-Cell FG, and Blastocyst SG vs. Blastocyst FG SM

Overall, 34 DEM were identified between the three SM comparison groups, in which 14 miRNAs belonged to 2-cell SG vs. 2-cell FG, 7 miRNAs were detected between 8-cell SG vs. 8-cell FG, and 13 miRNAs were differentially expressed between the blastocyst SG and blastocyst FG groups (Table 1). Of the 14 DEM detected between 2-cell SG and 2-cell FG, 12 miRNAs and 2 miRNAs were upregulated and downregulated (Table 2), respectively, in 2-cell SG SM in comparison to 2-cell FG SM. For 8-cell SG and 8-cell FG, 6 miRNAs and 1 miRNA were upregulated and downregulated (Table 3) respectively, in 8-cell SG SM in comparison to 8-cell FG SM. Of the 13 DEM detected between blastocyst SG and blastocyst FG SM, 9 miRNAs and 4 miRNAs were upregulated and downregulated (Table 4) in blastocyst SG SM in comparison to blastocyst FG SM, respectively. It appears that SG embryos are releasing, rather than up-taking, miRNAs in their environment. Differences in miRNA expression in the SM seem to be highest at the early and late stages of pre-implantation development between SG and FG embryos.

Interestingly, the majority of miRNAs detected in SM were expressed in a stage-specific manner, with only three miRNAs being co-detected in more than one SM condition. Bta-miR-1343-5p was upregulated in the SM of both 2-cell and 8-cell SG, in comparison to their 2-cell and 8-cell FG counterparts. Bta-miR-450b and bta-miR-760-5p were detected in both 8-cell and blastocyst SM. However, the expression of the two miRNAs differed between the two conditions. Both miRNAs were upregulated in the SM of 8-cell SG embryos, while the two miRNAs were downregulated in SM cultured with SG blastocyst. It should be noted that no miRNAs were consistently differentially expressed across all three SM groups.

### 3.2. Predictions of miRNA-mRNA Targets for Differentially Expressed miRNAs Detected between 2-Cell SG vs. 2-Cell FG, 8-Cell SG vs. 8-Cell FG, and Blastocyst SG vs. Blastocyst FG SM

With regards to 2-cell SG vs. 2-cell FG miRNAs, bta-miR-2361 did not have any predicted targets that met the cumulated weighted score cutoff and, thus, were not included in the analysis. From the remaining 13 miRNAs differentially expressed, 635 target mRNAs were predicted (Appendix A). One hundred thirty-seven mRNAs were significantly enriched across 35 biological pathways (Table 5). Proceeding to the 8-cell SG vs. 8-cell FG miRNAs, the analysis did not include bta-miR-2487 since it could not be located in the TargetScan database. From the remaining 6 DEM, 579 mRNA targets were predicted (Appendix A), and 141 mRNAs were significantly enriched across 36 different biological processes in DAVID (Table 6). Lastly, a total of 837 mRNA targets were predicted for the 12 miRNAs differentially expressed between the blastocyst SG vs. blastocyst FG condition (Appendix A). When inputted into DAVID, 239 mRNAs were enriched across 76 different biological processes (Table 7). Due to the significant number of biological processes enriched in each comparison group, only the top 5 pathways with the most genes enriched were featured on the tables. Overall, the majority of the predicted targets of DEM across the three conditions clustered around biological processes controlling transcription and proliferation.

Cross-referencing the enriched genes from each comparison group with their respective miRNA-mRNA gene-list allowed for the identification of miRNAs whose gene targets were enriched in DAVID. Among the 14 miRNAs that showed differential expression between 2-cell SG vs. 2-cell FG SM, DAVID exhibited a high representation of the bta-miR-1343-5p and bta-miR-2443 gene targets. Of the seven miRNAs detected between 8-cell SG vs. 8-cell FG SM, bta-miR-1343-5p and bta-miR-2885 gene targets had the highest representation in GSEA. Among the 12 miRNAs detected between blastocyst SG vs. blastocyst FG SM, the gene targets of bta-miR-6535 were overly represented in DAVID.

## 4. Discussion

This study is pioneering in its global profiling of miRNAs in the SM conditioned with embryos growing at different developmental rates. Our results indicate distinct miRNA populations can be identified between SG and FG embryos. More importantly, these unique miRNA signatures can be detected at the early, mid, and late stages of pre-implantation embryo development. Our results suggest SG embryos, at all conditions examined (2-cell SM, 8-cell SM, and blastocyst SM), preferentially release miRNAs into the extracellular environment. Although no other study has reported this, metabolomics studies have detected distinct metabolites in the SM conditioned with embryos differing in developmental rate and viability. These studies suggest lower-quality embryos are metabolically more active than their higher-quality counterparts. Since metabolism is influenced by the genes expressed within the cell, it can be postulated that increases in embryonic metabolism are preceded by higher genetic activity. Perhaps miRNAs are used within the embryo to initiate and modulate gene expression influencing metabolic turnover. Evidence does suggest the extracellular miRNA population serves as a good indicator of intracellular miRNA expression. Thus, our result indicates increases in metabolic activity in non-viable and SG embryos may also be driven by increases in miRNA expression, which are detectable in SM.

Across the three conditions examined, only the miRNAs detected between blastocyst SG vs. blastocyst FG SM had some previous annotation in literature. Specifically, miR-320a and miR-24-3p, which were detected to be upregulated in blastocyst SG SM, have also been cited in previous SM studies. According to Kropp and Khatib, miR-24-3p was one of the five miRNAs they detected to be upregulated in SM conditioned with degenerate blastocyst [8]. In a subsequent supplementation study, miR-24 was added to the plain media of morula stage embryos. Supplementation resulted in a 44-fold increase in expression of miR-24 in blastocyst cultured with miR-24 and a 27.3% decrease in blastocyst rates [8]. Kropp and Khatib postulate miR-24 influenced the expression of CDKN1b, which is a cell-cycle regulator. Despite the differences in the classification of non-viable embryos, whereby we used the developmental rate and Kropp and Khatib examined arrested/degenerate embryos, both studies indicate miR-24 may serve as a biomarker of embryo viability at the blastocyst stage of development.

Another miRNA identified in this study and previously annotated in literature was miR-320a. A recent study by Berkhout and colleagues suggest that miR-320a is a pre-implantation marker secreted by embryos. Specifically, the researchers profiled the miRNome of SM conditioned with embryos either scoring low or high in morphological scores [25]. It was determined that miR-320a was secreted by higher-quality embryos. Subsequently, miR-320a was supplemented in the culture media of human embryonic stem cells. Berkhout and colleagues reported miR-320a was able to stimulate the migration of decidualized human embryonic stem cells, with a downstream transcriptome analysis revealing miR-320a modulates genes regulating cell adhesion and cytoskeleton organization [25]. Interestingly, our study found miR-320a to be upregulated in SM conditioned with SG embryos. Capalbo and colleagues have postulated embryos release miRNAs into the extracellular environment as a means of paracrine communication with endometrial tissue [6]. Thus, the findings in our study suggest miR-320a may serve to inhibit implantation as it was found to be released by SG embryos. Although consensus about the role of miR-320a is mixed, it should be noted that our study and the one conducted by Berkhout and colleagues were carried out in different species and culture conditions were not identical. Thus, inter-species differences and environmental conditions may have influenced the findings of both studies. Perhaps, miR-320a may have both inhibitory and stimulatory effects on implantation as miRNAs have various targets within the genome.

Aside from miR-320a and miR-24, miR-615 and miR-17 have also been previously annotated in literature, albeit in cancer-related studies. Specifically, miR-615 has been characterized to play a role in angiogenic events influencing tumorigenesis. Icli and colleagues demonstrated that miR-615 has anti-angiogenic effects, whereby the expression of the miRNA significantly inhibited endothelial cell proliferation and migration [26]. Similarly, miR-17 has been cited in literature to have anti-oncogenic effects. Hossain and colleagues reported miR-17 transfection in breast cancer tissue resulted in the translational repression of the breast-cancer-associated gene *A1B1* [27]. The subsequent downregulation of *A1B1* decreased breast cancer proliferation. Thus, it seems the miRNAs upregulated in SG embryos at the blastocyst stage have roles in cancer development. Cancer and embryogenesis rely on similar pathways for growth and development. Therefore, it is interesting to see that cancer-related miRNAs are detectable in media conditioned with embryos growing at different rates. Perhaps these anti-proliferative miRNAs in cancer serve to inhibit growth and development in an embryo.

Aside from previously annotated miRNAs, GSEA analysis also revealed novel stage-specific miRNA biomarkers. Bta-miR-1343 and bta-miR-2443 were miRNAs upregulated in SM cultured with 2-cell SG embryos. GSEA analysis indicated the gene targets of the two miRNAs had roles in regulating transcription and cell proliferation. Specifically, the majority of gene targets had biological implications pertaining to the positive and negative regulation of transcription from RNA polymerase II promoter. This is an interesting finding as previous research suggests little to no transcription occurs at the 2-cell stage in bovine embryos. Prior to the 8-cell stage, the developing bovine embryo relies on parentally inherited transcripts (mRNA and miRNA) and proteins for survival. Therefore, it is unclear why the gene targets of miR-1343 and miR-2443 would cluster around promoting transcriptional events.

However, research by Vassena and colleagues does suggest embryos are capable of transcription prior to embryonic genome activation (EGA). Through the genomic-wide transcript analysis of human oocytes and embryos, Vassena and colleagues detected a series of successive waves of embryonic transcriptional initiation events beginning as early as the 2-cell stage [28]. Therefore, their findings suggest transcriptional events in human embryos may begin as early as the 2-cell stage, and not at the 4–8 cell stage as previously thought. Although unexplored in bovine embryos, our results indicate transcriptional events may be occurring earlier than EGA. It can be hypothesized SG embryos may initiate transcriptional events earlier as a response to its delayed development. The early activation of the embryonic genome may serve as a repair mechanism to salvage an embryo during the pre-implantation period.

It should also be noted bta-miR-1343 and bta-miR-2443 had gene targets relating to spermatogenesis. Previous research profiling the origins of embryonic miRNAs has suggested the majority of miRNAs expressed prior to EGA are of maternal origin. However, researchers did discover that embryos can also inherit sperm-borne miRNAs. Although bta-miR-1343 and bta-miR-2443 have not been annotated in mammalian sperm, results from our study suggest that these miRNAs are of paternal origin, capable of influencing transcriptional events in an embryo.

It should also be highlighted that bta-miR-1343, bta-miR-760-5p, and bta-miR-450b were co-detected in more than one SM condition. Bta-miR-1343 was found to be upregulated in the SM of SG embryos at the 2-cell and 8-cell stage. With regard to bta-miR-760-5p and bta-miR-450bs, their expression was detected in media conditioned with the SG 8-cell and blastocyst embryos. Contrasting bta-miR-1343, both bta-miR-760-5p and bta-miR-450b were upregulated in 8-cell SG media, then were downregulated in blastocyst SG media. Although all three miRNAs have not been previously annotated in embryos or in SM, their consistent expression between 2-cell and 8-cell or 8-cell and blastocyst SM, respectively, suggest they may have functional roles in normal and aberrant embryo development.

## 5. Conclusions

Overall, this study was the first to detect miRNA expression differences in the SM between SG and FG embryos with miRNAomic analysis. Across all three developmental stages examined (2-cell, 8-cell, and blastocyst), embryos with delayed development expressed more miRNAs in the SM than those developing at a faster rate. It is postulated that the difference in miRNA expression is associated with increases in metabolic activity observable in non-viable embryos. Moreover, our findings also highlight novel miRNA biomarkers correlated with slow- and fast-growing embryos at the 2-cell, 8-cell, and blastocyst stage of development. Future research should focus on validating these miRNAs in the SM and within the embryo with subsequent gene expression studies to further elucidate the roles of these miRNAs in embryonic development.

## Figures and Tables

**Figure 1 animals-14-02331-f001:**
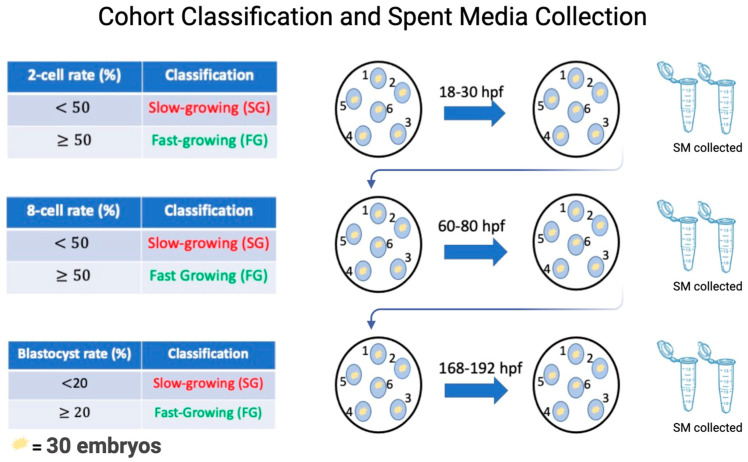
To visualize this process, at day 0 of culture, 30 embryos were placed in a microdrop of IVC media and assigned a number from 1–6. Once this was established, the embryos were cultured to the 2-cell stage. At the 2-cell stage, each group was morphologically assessed from 2-cell rate formation, whereby groups with less than 50% reaching 2-cell stage being labelled as slow-growing and groups with more than or equal to 50% 2-cell rate being considered fast-growing. Once this was recorded, the embryos were transferred to a fresh microdrop of IVC media and allowed to culture to the 8-cell stage. The respective spent media from 2-cell slow- and fast-growing embryos were pooled together in separate Eppendorf tubes and flash-frozen until RNA extraction. Once at the 8-cell stage, morphological assessment was conducted again to assess 8-cell rate formation. Similarly, groups with less than 50% 8-cells were considered slow-growing and groups with more than or equal to 50% fast-growing. Once recorded, the same cohort of embryos were transferred to the fresh drop of IVC media and cultured to the blastocyst stage. SM from slow- and fast-growing 8-cell embryos were collected. Once at the blastocyst stage, morphological assessment was conducted for the last time and groups with less than 20% blastocyst were considered slow-growing and more than or equal to 20% fast-growing. Once recorded, the embryos and the respective media were collected and frozen. The experiment was repeated 8 times.

**Table 1 animals-14-02331-t001:** Differentially expressed miRNAs (DEM) between 2-cell SG vs. 2-cell FG, 8-cell SG vs. 8-cell FG, and blastocyst SG vs. blastocyst FG SM: total, upregulated, and downregulated.

Cell Stage	Number of DEM between SG/FG SM	Upregulated DEM at This Stage	Downregulated DEM at This Stage
2-cell	14	12	2
8-cell	7	6	1
Blastocyst	13	9	14

**Table 2 animals-14-02331-t002:** DEM between 2-cell SG SM vs. 2-cell FG SM: downregulated (red) and upregulated (green). The majority of miRNAs were upregulated in 2-cell SG SM in comparison to 2-cell FG SM.

miRNAs	Fold-Change	p-Value
bta-miR-455-3p	−2.83	1.19 × 10^−6^
bta-miR-628	−2.07	0.0104
bta-miR-2359	2.02	0.0113
bta-miR-2412	2.03	0.0025
bta-miR-2452	2.68	0.0006
bta-miR-2325a	3.12	0.003
bta-miR-1343-5p	3.16	0.0184
bta-miR-2421	4.04	0.0002
bta-miR-2434	6.56	0.0005
bta-miR-2393	13.36	0.0004
bta-miR-2444	17.58	0.0005
bta-miR-2361	41.56	0.001
bta-miR-3613	47.62	0.0002
bta-miR-2325c	58.04	0.0003

**Table 3 animals-14-02331-t003:** DEM between 8-cell SG SM vs. 8-cell FG SM: downregulated (red) and upregulated (green). The majority of miRNAs were upregulated in 8-cell SG SM in comparison to 8-cell FG SM.

miRNAs	Fold-Change	p-Value
bta-miR-3613b	−6.1	9.48 × 10^−6^
bta-miR-1343-5p	2.14	0.0149
bta-miR-450b	3.04	0.0002
bta-miR-2487	3.23	6.61 × 10^−6^
bta-miR-2885	4.09	1.48 × 10^−7^
bta-miR-1281	4.27	0.0033
bta-miR-760-5p	4.55	0.0001

**Table 4 animals-14-02331-t004:** DEM between blastocyst SG SM vs. blastocyst FG SM: downregulated (red) and upregulated (green). The majority of miRNAs were upregulated in SG SM in comparison to FG SM.

miRNAs	Fold-Change	p-Value
bta-miR-450b	−2.71	0.0002
bta-miR-760-5p	−2.48	0.0076
bta-miR-2296	−2.34	0.0004
bta-miR-6535	−2.11	0.007
bta-let-7b	2.05	1.26 × 10^−5^
bta-miR-2402	2.18	1.66 × 10^−5^
bta-miR-23a	2.34	5.21 × 10^−7^
bta-miR-23b-3p	2.45	3.90 × 10^−6^
bta-miR-17-5p	2.47	0.0102
bta-miR-2898	2.51	0.0046
bta-miR-615	2.66	0.0035
bta-miR-320a	2.79	1.45 × 10^−8^
bta-miR-24-3p	2.83	8.65 × 10^−7^

**Table 5 animals-14-02331-t005:** Top 5 enriched biological pathways of predicted genes regulated by miRNAs differentially expressed in 2-cell SG SM vs. 2-cell FG SM.

GO Term	Genes	p-Value
Positive regulation of transcription from RNA polymerase II promoter	33	3.05 × 10^−4^
Negative regulation of transcription from RNA polymerase II promoter	26	4.65 × 10^−4^
Negative regulation of transcription, DNA-templated	14	0.03335617
Negative regulation of cell proliferation	13	0.02255407
Spermatogenesis	12	0.01965493

**Table 6 animals-14-02331-t006:** Top 5 enriched biological pathways of predicted genes regulated by miRNAs differentially expressed in 8-cell SG SM vs. 8-cell FG SM.

GO Term	Genes	p-Value
Positive regulation of transcription from RNA polymerase II promoter	33	1.86 × 10^−4^
Intracellular signal transduction	14	0.045985
Negative regulation of cell proliferation	13	0.01862582
Protein transport	11	0.03235415
Positive regulation of ERK1 and ERK2 cascade	10	0.01051371

**Table 7 animals-14-02331-t007:** Top 5 enriched biological pathways of predicted genes regulated by miRNAs differentially expressed in blastocyst SG SM vs. blastocyst FG SM.

GO Term	Genes	p-Value
Positive regulation of transcription from RNA polymerase II promoter	36	0.0078327
Regulation of transcription from RNA polymerase II promoter	26	1.16 × 10^−5^
Positive regulation of cell proliferation	20	0.0062254
Small GTPase mediated signal transduction	19	8.66 × 10^−4^
Positive regulation of transcription, DNA-templated	18	0.0236799

## Data Availability

Data are contained within the article and Appendix A.

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
