# Peer review of "MicroRNAomic Analysis of Spent Media from Slow- and Fast-Growing Bovine Embryos Reveal Distinct Differences"

_animals, 2024, doi:10.3390/ani14162331_

Round 1

Reviewer 1 Report

Comments and Suggestions for Authors

In the original article MicroRNAomic analysis of spent media from slow and fast-growing bovine embryos reveal distinct differences submitted by Paul Del Rio to Animals, the authors devided the early embryos as slow or fast-growing embryos based on the percentage of embryos that had reached the desired morphological stage and identified the differentially expressed miRNAs between spent media of slow and fast-growing embryos. I have a few comments and suggestions

1) Introduction

Line 94-95: The author mentioned Recent data from our lab has also shown the miRNA expression in the SM is different between SG and FG embryos. What are the differences between previous study and this study?

In addition to their own studies, authors should provide evidence that miRNAs can get into spent media.

2) Materials and Methods

There are many errors in superscript and subscript.

3) Line 176: The authors mentioned 1100μL of SM. How to achieve it? Please explain in detail.

4) Results

Whether KEGG analysis of differentially expressed miRNA-targeting genes was performed? The description of the result is too simple. The authors also did not verify the differentially expressed miRNAs.

5) The English language needs moderate editing.

Comments on the Quality of English Language

The English language needs moderate editing.

Author Response

1) Introduction

Line 94-95: The author mentioned “Recent data from our lab has also shown the miRNA expression in the SM is different between SG and FG embryos”. What are the differences between previous study and this study?

These sentences were from our other work and have now been removed from the paper. We apologize for the confusion. 

In addition to their own studies, authors should provide evidence that miRNAs can get into spent media.

Several references in the discussion have been added to provide evidence of miRNA being present in the spent media. For example, ref. 8, 27,  28, 31 all demonstrate expression of miRNA in the spent media. 

2) Materials and Methods

There are many errors in superscript and subscript.

Thank you for the suggestion. We have corrected the anomalies.

3) Line 176: The authors mentioned “1100μL of SM”. How to achieve it? Please explain in detail.

25μl of conditioned media from each microdrop from SG and FG groups were collected, pooled, and placed separately in Eppendorf tubes. Multiple IVF runs were completed until 1100μl of SM was collected for each developmental stage: 2-cell SG/2-cell FG, 8-cell SG/8-cell FG, and blastocyst SG/blastocyst SG.

4) Results

Whether KEGG analysis of differentially expressed miRNA-targeting genes was performed? The description of the result is too simple. The authors also did not verify the differentially expressed miRNAs.

It is true that the KEGG analysis was done and differentially expressed miRNAs were validated however those become part of our next study which will be published soon. The content of this paper was becoming too long with addition of KEGG analysis and the confirmation studies. 

5) The English language needs moderate editing.

Thank you! We have sought professional editing help and corrections have been made accordingly (marked in green). 

Reviewer 2 Report

Comments and Suggestions for Authors

I have gone through the manuscript. The results of this study will contribute to reveal biomarkers related to intrinsic embryo physiology, and I would like to reconsider after major revision.

Line 135: Is the method of denudation by vortex a common practice? Is there a possibility of cell damage? Additionally, please indicate that the cumulus cells have been sufficiently removed and that there is no possibility of contamination with miRNA from cumulus cells in subsequent experiments.

Line 151-177: The series of embryo operations and collection of culture media is a little difficult to understand, so please create a work flow diagram.

Line 178-216, Table 1-3: Please describe the number of samples that have undergone miRNA extraction, microarray hybridization, and statistical analysis in your paper.

Line 229: Before the results in Table 1-3, please provide a table of basic statistics (e.g., percentage of embryonic stages for each group, the number of samples, average, standard deviation, and so on). If the standard deviation is large, outliers should be excluded.

Line 277: One hundred thirty seven mRNAs

Line 329-330: Please describe in detail the evidence suggesting that extracellular miRNA populations serve as excellent indicators of intracellular miRNA expression, citing previous publications.

Line 330-331: Is the miRNA released by non-viable embryos and SG cells related to exosomes and cell-to-cell communication? If you have a relevant report, please cite and explain.

Author Response

Line 135: Is the method of denudation by vortex a common practice? Is there a possibility of cell damage? Additionally, please indicate that the cumulus cells have been sufficiently removed and that there is no possibility of contamination with miRNA from cumulus cells in subsequent experiments.

Use of vortex is a standard process for removing cumulus cells (https://animal.ifas.ufl.edu/media/animalifasufledu/hansen-lab-website/lab-protocols/Laboratory-Guide-for-Production-of-Bovine-Embryos-In-Vitro-ver-03.15.2023.pdf). One of the bovine IVF protocols as outlined by Dr. Pete Hansen's lab clearly shows the process of vortexing. 

The sentence indicating no contamination from cumulus cells happened during in vitro culture has been added to the paper (Line 157-158)

Line 151-177: The series of embryo operations and collection of culture media is a little difficult to understand, so please create a work flow diagram.

Thank you for the suggestion. The flow diagram is attached as Figure 1 (Line 160)

Line 178-216, Table 1-3: Please describe the number of samples that have undergone miRNA extraction, microarray hybridization, and statistical analysis in your paper.

In total, 3 biological replicates of pooled SM from 2-cell SG/2-cell FG, 8-cell SG/8-cell FG, and blastocyst SG/blastocyst FG groups and plain media was processed and prepared for microarray analysis (Line 199 and 201). Statistical analysis has been outlined in lines 214 to 224. 

Line 229: Before the results in Table 1-3, please provide a table of basic statistics (e.g., percentage of embryonic stages for each group, the number of samples, average, standard deviation, and so on). If the standard deviation is large, outliers should be excluded.

As in our previously published paper (https://www.frontiersin.org/articles/10.3389/fvets.2021.658968/full) the standard IVF production information was provided. On average, our cleavage rate was about 92% and blastocyst rates were about 30%. Outliers with low cleavage rates were excluded. Three biological replicates were used to collected pooled SM from 2-cell SG/2-cell FG, 8-cell SG/8-cell FG, and blastocyst SG/blastocyst FG groups and plain media. 

Line 277: One hundred thirty seven mRNAs

Thank you for the suggestion. The change has been made. 

Line 329-330: Please describe in detail the evidence suggesting that extracellular miRNA populations serve as excellent indicators of intracellular miRNA expression, citing previous publications.

Several references have been added in the discussion highlighting the suggestions made. For example, references 8, 27, 28, 29, 30 and 31.

Line 330-331: Is the miRNA released by non-viable embryos and SG cells related to exosomes and cell-to-cell communication? If you have a relevant report, please cite and explain.

Aside from miR-320a and miR-24, miR-615 and miR-17 have also been previously annotated in literature, albeit in cancer-related studies. Specifically, miR-615 has been characterized to play a role in angiogenic events influencing tumorigenesis. Icli and colleagues demonstrated that miR-615 has anti-angiogenic effects, whereby expression of the miRNA significantly inhibited endothelial cell proliferation and migration [29]. Similarly, miR-17 has been cited in literature to have anti-oncogenic effects. Hossain and colleagues reported miR-17 transfection in breast cancer tissue resulted in the translational repression of the breast cancer associated gene A1B1 [30]. The subsequent downregulation of A1B1 decreased breast cancer proliferation. Thus, it seems the miRNAs upregulated in SG embryos at the blastocyst stage have roles in cancer development. Cancer and embryogenesis rely on similar pathways for growth and development. Therefore, it is interesting to see that cancer-related miRNAs are detectable in media conditioned with embryos growing at different rates. Perhaps, these anti-proliferative miRNAs in cancer, serve to inhibit growth and development in an embryo.

Figure 1 Legend:

Figure 1: To visualize this process, at day 0 of culture, 20 embryos were placed in a microdrop of IVC media and assigned a number from 1-6. Once this was established, the embryos were cultured to the 2-cell stage. At the 2-cell stage, each group was morphologically assessed from 2-cell rate formation, whereby groups with less than 50% reaching 2-cell stage being labelled as slow growing and groups with more than or equal to 50% 2-cell rate being considered fast growing. Once this was recorded, the embryos were transferred to a fresh microdrop of IVC media and allowed to culture to the 8-cell stage. The respective spent media from 2-cell slow and fast growing embryos were pooled together in separate eppendof tubes and flash frozen until RNA extraction. Once at the 8-cell stage, morphological assessment was conducted again to assess 8-cell rate formation. Similarly, groups with less than 50% 8-cells were considered slow growing and groups with more than or equal to 50% fast growing. Once recorded, the same cohort of embryos were transferred to the fresh drop of IVCmedia and cultured to the blastocyst stage. SM from slow and fast growing 8-cell embryos were collected. Once at the blastocyst stage, morphologvcial assessment was conducted for the last time and groups with less than 20% blastocyst were condisered slow growing and more than or equal to 20% fast growing. Once recorded, the embryos and the respective media was collected and frozen.

Reviewer 3 Report

Comments and Suggestions for Authors

General comments:

This study presents interesting data on the expression of miRNAs throughout different stages of embryonic development, making pertinent contributions to the subject. The writing of the manuscript is adequate and relevant to what is proposed, there were a few problems that I observed during my review, and after revisions of these, I believe that the article will be suitable for publication. I would also like to suggest an English proofreading, due to some small spelling mistakes that I did not find relevant to highlight at this first moment. Below, I present details of the review.

Specific comments:

Abstract

Lines 20 and 22: Remove the “-“ in “in-vitro”.

Keywords: Replace the keywords that are already in the title of the manuscript, such as spent media and bovine.

Introduction

Line 41: Define the abbreviation “IVF”.

Lines 52, 53, 54, 61, 63, 65: Remove the “-“ in “in vivo” and “in-vitro”.

Line 72: Define the abbreviation “NMR”.

Line 94: The way you express yourself is a bit informal. Modify the sentence. For example, “Recent data from Laboratorio “lab name” has shown that miRNA expression in the SM is different between SG and FG embryos.”

Line 96: Add a comma after “miR-370”.

Line 98: Add a comma after “miR-155-5p”.

Material and methods

Lines 105-108: What is the document number with the authorization?

Lines 109-11: This topic is not necessary, it can be removed.

Line 112: Remove the “-“ in “in-vitro”.

Lines 112-150: In the topic “Oocyte collection and in-vitro production of bovine embryos”, is the methodology described your own or was it based on any method described? If it is based on a described method, please cite the author and give the reference.

Line 151: Remove the “-“ in “in-vitro”.

Line 156: Change “a” by “an”.

Lines 151-192: In the topics “Collection of spent in-vitro culture media conditioned with SG and FG embryos” and “miRNA extraction”, is the methodology described your own, or was it based on any method described? If it is based on a described method, please cite the author and give the reference.

Line 191: Change “was” by “were”.

Line 219: Change “was” by “were”.

Results, discussion, and conclusions

The topics are appropriate and coherent. Only grammatical revision is needed.

Comments on the Quality of English Language

The main problem with the quality of English is the use of grammatical symbols such as comma and dash.

Author Response

Abstract

Lines 20 and 22: Remove the “-“ in “in-vitro”.

The correction has been made in the entire paper.

Keywords: Replace the keywords that are already in the title of the manuscript, such as spent media and bovine.

Thank you for the suggestion. The changes have been made. 

Introduction

Line 41: Define the abbreviation “IVF”. 

The change is reflected in the paper now. 

Lines 52, 53, 54, 61, 63, 65: Remove the “-“ in “in vivo” and “in-vitro”.

The correction has been made in the entire paper.

Line 72: Define the abbreviation “NMR”. 

The correction has been made.

Line 94: The way you express yourself is a bit informal. Modify the sentence. For example, “Recent data from Laboratorio “lab name” has shown that miRNA expression in the SM is different between SG and FG embryos.”

The changes have been made. 

Line 96: Add a comma after “miR-370”. 

This section has been deleted.

Line 98: Add a comma after “miR-155-5p”.

This section has been deleted.

Material and methods

Lines 105-108: What is the document number with the authorization?

No authorization was required as the work was done on ovaries collected from slaughterhouse. This sentence has been added to the paper. 

Lines 109-11: This topic is not necessary, it can be removed.

The journal requires this statement. 

Line 112: Remove the “-“ in “in-vitro”.

The changes have been made. 

Lines 112-150: In the topic “Oocyte collection and in-vitro production of bovine embryos”, is the methodology described your own or was it based on any method described? If it is based on a described method, please cite the author and give the reference.

The references have been added. 

Line 151: Remove the “-“ in “in-vitro”.

Done!

Line 156: Change “a” by “an”.

Done!

Lines 151-192: In the topics “Collection of spent in-vitro culture media conditioned with SG and FG embryos” and “miRNA extraction”, is the methodology described your own, or was it based on any method described? If it is based on a described method, please cite the author and give the reference.

The references have been added (21, 24, 25). 

Line 191: Change “was” by “were”.

English editor prefers use of was

Line 219: Change “was” by “were”.

English editor prefers use of was

Results, discussion, and conclusions

The topics are appropriate and coherent. Only grammatical revision is needed.

Thank you for your suggestions and in helping us improve the paper!

Round 2

Reviewer 1 Report

Comments and Suggestions for Authors

The author basically made the modification according to the revision suggestions. But I still think that the results of GO and KEGG analysis should be put together, which can obtain the comprehensive understanding of the differentially expressed miRNAs. Functional verification of differentially expressed miRNAs can be published in the next study.

Comments on the Quality of English Language

The English language of the article has been significantly improved. And there are small mistakes, please revise.

Author Response

The author basically made the modification according to the revision suggestions. But I still think that the results of GO and KEGG analysis should be put together, which can obtain the comprehensive understanding of the differentially expressed miRNAs. Functional verification of differentially expressed miRNAs can be published in the next study.

We understand and appreciate your feedback. We believe we included the most pertinent details necessary for this study's objectives. The results of the KEGG analysis and the verification studies will be presented in a subsequent manuscript, which is already prepared for publication. We believe this approach ensures clarity and focus in our current study while allowing a detailed exploration in the follow-up publication.

Reviewer 2 Report

Comments and Suggestions for Authors

Line 162: 30 PZ ? In Figure 1, it says 20 embryos.

Figure 1: The flow diagram is  difficult to understand, so please recreate. I'm not sure what you mean by "same cohort of embryos per ran”. How many times was this experiment repeated, how was it averaged, and how was it statistically analyzed?

Line 234: Before the results in Table 1-3, please provide a table of basic statistics (e.g., percentage of embryonic stages for each droplet, the number of samples, average, standard deviation, and so on).  I'm not sure what you mean by "Outliers with low cleavage rates were excluded" in the author responce. In other words, make sure Tables 1, 2, and 3 contain enough information. 

Author Response

Line 162: 30 PZ ? In Figure 1, it says 20 embryos.

Each microdroplet contained 30 PZ. The Figure has been updated accordingly.

Figure 1: The flow diagram is  difficult to understand, so please recreate. I'm not sure what you mean by "same cohort of embryos per ran”. How many times was this experiment repeated, how was it averaged, and how was it statistically analyzed?

Thank you for the feedback, the flow diagram has been recreated. “Same cohort of embryos per run” was revised to follow the same cohort of embryos throughout the pre-implantation period, for clarity. A full legend is provided. The experiment was repeated 8 times, until 1100μl of SM was collected for each developmental stage: 2-cell SG/2-cell FG, 8-cell SG/8-cell FG, and blastocyst SG/blastocyst SG.

Line 234: Before the results in Table 1-3, please provide a table of basic statistics (e.g., percentage of embryonic stages for each droplet, the number of samples, average, standard deviation, and so on).  I'm not sure what you mean by "Outliers with low cleavage rates were excluded" in the author responce. In other words, make sure Tables 1, 2, and 3 contain enough information. 

Table 1 was created to show the total number of differentially expressed miRNAs (DEM), upregulated DEM, and downregulated DEM between 2-cell SG vs. 2-cell FG, 8-cell SG vs. 8-cell FG, and blastocyst SG vs. blastocyst FG SM. The table depicts what was written in the results to provide context for Tables 2-4.

Reviewer 3 Report

Comments and Suggestions for Authors

Once the suggested revision has been carried out, the manuscript is ready for publication.

Author Response

Thank you for your suggestions and in helping us improve the paper!